# Genome-Wide Association Study of Maternal Genetic Effects on Intramuscular Fat and Fatty Acid Composition in Rabbits

**DOI:** 10.3390/ani13193071

**Published:** 2023-09-30

**Authors:** Ayman G. EL Nagar, Imen Heddi, Bolívar Samuel Sosa-Madrid, Agustín Blasco, Pilar Hernández, Noelia Ibáñez-Escriche

**Affiliations:** 1Institute for Animal Science and Technology, Universitat Politècnica de València, 46022 Valencia, Spain; ayman.elnagar@fagr.bu.edu.eg (A.G.E.N.);; 2Department of Animal Production, Faculty of Agriculture at Moshtohor, Benha University, Benha 13736, Egypt; 3Centro Regional de Selección y Reproducción Animal (CERSYRA), Av. del Vino, 10, 13300 Valdepeñas, Spain

**Keywords:** divergent selection, maternal genetic effects, intramuscular fat, fatty acids, GWAS, rabbits

## Abstract

**Simple Summary:**

Ten generations of divergent selection for intramuscular fat (IMF) content was performed in rabbits. Through the generations, the IMF content has improved. Different Bayesian animal models were used to assess the importance of maternal genetic effects on the genetic control of IMF and fatty acid composition of the two rabbit lines divergently selected for IMF content. Additionally, maternal genome-wide association studies were carried out in these lines to identify genomic regions affecting the IMF and intramuscular fatty acid composition of the offspring. The maternal genetic variance explained an important part of the phenotypic variance of the IMF and intramuscular fatty acid composition. Potential candidate genes were identified on the associated genomic regions related to the traits under study.

**Abstract:**

Maternal genetic effects (MGE) could affect meat quality traits such as intramuscular fat (IMF) and its fatty acid composition. However, it has been scarcely studied, especially in rabbits. The objectives of the present study were, first, to assess the importance of MGE on intramuscular fat and fatty acid composition by applying a Bayesian maternal animal model in two rabbit lines divergently selected for IMF. The second objective was to identify genomic regions and candidate genes of MGE that are associated with the traits of these offspring, using Bayesian methods in a Genome Wide Association Study (GWAS). Quantitative analyses were performed using data from 1982 rabbits, and 349 animals from the 9th generation and 76 dams of the 8th generation with 88,512 SNPs were used for the GWAS. The studied traits were IMF, saturated fatty acids (total SFA, C14:0; myristic acid, C16:0; palmitic acid and C18:0; stearic acid), monounsaturated fatty acids (total MUFA, C16:1n-7; palmitoleic acid and C18:1n-9; oleic acid), polyunsaturated fatty acids (total PUFA, C18:2n-6; linoleic acid, C18:3n-3; α-linolenic acid and C20:4n-6; arachidonic acid), MUFA/SFA and PUFA/SFA. The proportion of phenotypic variance explained by the maternal genetic effect ranged from 8 to 22% for IMF, depending on the model. For fatty acid composition, the proportion of phenotypic variance explained by maternal genetic effects varied from 10% (C18:0) to 46% (MUFA) in a model including both direct and additive maternal genetic effects, together with the common litter effect as a random variable. In particular, there were significant direct maternal genetic correlations for C16:0, C18:1n9, C18:2n6, SFA, MUFA, and PUFA with values ranging from −0.53 to −0.89. Relevant associated genomic regions were located on the rabbit chromosomes (OCU) OCU1, OCU5 and OCU19 containing some relevant candidates (*TANC2*, *ACE*, *MAP3K3*, *TEX2*, *PRKCA*, *SH3GL2*, *CNTLN*, *RPGRIP1L* and *FTO*) related to lipid metabolism, binding, and obesity. These regions explained about 1.2 to 13.9% of the total genomic variance of the traits studied. Our results showed an important maternal genetic effect on IMF and its fatty acid composition in rabbits and identified promising candidate genes associated with these traits.

## 1. Introduction

Many traits of interest in livestock production are affected by the mother of the individual in whom the trait is observed [1]. Maternal effects are defined by [2] as the causal influence of the maternal genotype (maternal genetic effects; MGE) or phenotype (maternal environmental effects) on the offspring phenotype. For instance, the intrauterine environment, dam’s milk production and the mothering ability represent maternal effects, which may be a combination of genetic and environmental factors that influence the offspring [3]. Maternal effects constitute a sizeable source of variation in traits of young mammals [3]. In rabbits and pigs, the importance of maternal effects for traits recorded in the fattening period is a consequence of the short interval of time between weaning and slaughtering. Additionally, the maternal effect might also influence post-weaning growth as a carry-over effect from weaning weight [4]. Similarly, the importance of the litter effect on intramuscular fat (IMF) content and its fatty acid composition has been confirmed in fat pigs such as Iberian [5] and Duroc [6]. However, these studies did not distinguish between the MGE and maternal environment or litter effect. Intramuscular fat content and its fatty acid composition are of interest in meat-producing animals since they are directly related to the quality attributes of meats, including organoleptic properties, fat consistency, shelf life, nutritional value and health issues [7,8,9]. The direct genetic control of IMF and its fatty acid composition has been determined in rabbits [10,11] and pigs [12,13,14] by both classical quantitative genetics and genome-wide association studies. However, to our knowledge, studies concerning the maternal genetic effects on IMF and fatty acid composition in livestock, particularly rabbits, are scarce. Thus, the main objectives of the present study were: (1) to quantify the importance of maternal genetic effects in the estimation of variance components for IMF and fatty acid composition using different Bayesian animal models and to determine the most appropriate model, (2) to identify genomic regions and potential candidate genes for maternal genetic effects associated with the traits studied in two rabbit lines divergently selected for IMF.

## 2. Materials and Methods

### 2.1. Ethical Statement

Animal management and the experimental procedures were approved by the Ethical Committee of the Universitat Politècnica de València, according to Council Directives 98/58/EC and 2010/63/EU, (reference number 2017/VSC/PEA/00212).

### 2.2. Animals and Phenotypes

The data used in this study were obtained from two rabbit lines involved in a divergent selection experiment aimed at modifying the intramuscular fat (IMF) content in the Longissimus thoracis et lumborum (LTL) muscle. This experiment was carried out over 10 generations (during the period from 2011 to 2019) in a rabbit selection nucleus located at the farm of the Institute of Animal Science and Technology of the Universitat Politècnica de València, Valencia, Spain. The base population consisted of 13 sires and 83 dams. Up to the 7th generation, each line consisted of 8 sires and 40 dams per generation. From the 8th generation on, the number increased to 10 sires and 60 dams per line. The selection was based on the offspring phenotypic value of IMF. For each dam, two full sibs (one male and one female) of first-parity offspring were sacrificed at nine weeks of age, and their IMF values were measured. The dams were then ranked according to these measurements. The top twenty percent of dams, as determined by this ranking, were selected to provide all the females for the next generation. Each sire was mated with five dams, and, in order to minimize the inbreeding, only one male offspring of the sire from the highest-ranking mating was selected for breeding the next generation. Further information on the divergent selection experiment can be found in previous publications [15,16]. The present study was carried out with data from 1982 rabbits, 166 of which were from the base population, 874 from the High-IMF line (H) and 942 from the Low-IMF line (L). The pedigree used had 3677 animals. Both lines (H and L) were contemporarily reared in the same rabbitry under the same environmental and management conditions, and fed ad libitum with a standard commercial diet, containing 15.1% crude protein, 14.5% crude fiber and 2.8% fat.

Rabbits were slaughtered at 9 weeks of age by exsanguination after electrical stunning. Carcasses were stored at 4 °C for 24 h, and then LTL muscle was excised from the carcass. LTL muscle minced and lyophilized. The IMF content of 1st to 8th generation samples were determined by near-infrared reflectance spectroscopy (NIRS) using the calibration equation developed by [17]. Twenty percent of 1st to 8th generation samples were chemically analysed by ether extraction after acid hydrolysis to validate the NIRS results. Chemical analysis was performed on all 8th and 9th generation LTL samples. IMF content was quantified as grams of IMF per 100 g of fresh muscle. Fatty acid methyl esters (Fame) were prepared as described by [18] and analyzed in a Focus Gas Chromatograph. More details of the experiment can be found in a previous publication [19]. IMF and fatty acid content were measured in grams per 100 g of fresh muscle, with fatty acids expressed as a percentage of total fatty acids. The fatty acids analyzed included the major individual fatty acids C14:0, C16:0, C18:0, C16:1n-7, C18:1n-9, C18:2n-6, C18:3n-3 and C20:4n-6 and the categories total SFA (saturated fatty acids), MUFA (monounsaturated fatty acids) and PUFA and the ratios MUFA/SFA and PUFA/SFA.

### 2.3. Quantifying the Importance of Maternal Genetic Effects

Five bivariate Bayesian animal models [20] of increasing complexity were implemented to estimate variance components and consequently assess the relevance of maternal genetic effects on IMF and their fatty acids. The variance components of IMF were estimated with univariate models, whereas bivariate models, always including the IMF trait to account for the selection effect, were fitted for fatty acids. All models and traits had the same fixed effects: month of slaughtering (51 levels), sex (2 levels: male and female), parity order (2 levels: 1st parity, ≥2nd parity), and method of fatty acid measurement (2 levels: NIRS, chemistry analysis). The five following bivariate models included different random effects:

Model 1. Animal model with direct additive genetic effects:$$\left[\begin{array}{c} y_{1}\\ y_{2} \end{array}\right]=\left[\begin{array}{cc} X_{1} & 0\\ 0 & X_{2} \end{array}\right]\left[\begin{array}{c} b_{1}\\ b_{2} \end{array}\right]+\left[\begin{array}{cc} Z_{d1} & 0\\ 0 & Z_{d2} \end{array}\right]\left[\begin{array}{c} a_{d1}\\ a_{d2} \end{array}\right]+\left[\begin{array}{c} e_{1}\\ e_{2} \end{array}\right]$$

Model 2. Animal model with direct additive genetic effects and maternal genetic effects:$$\left[\begin{array}{c} y_{1}\\ y_{2} \end{array}\right]=\left[\begin{array}{cc} X_{1} & 0\\ 0 & X_{2} \end{array}\right]\left[\begin{array}{c} b_{1}\\ b_{2} \end{array}\right]+\left[\begin{array}{cc} Z_{d1} & 0\\ 0 & Z_{d2} \end{array}\right]\left[\begin{array}{c} a_{d1}\\ a_{d2} \end{array}\right]+\left[\begin{array}{cc} Z_{m1} & 0\\ 0 & Z_{m2} \end{array}\right]\left[\begin{array}{c} a_{m1}\\ a_{m2} \end{array}\right]+\left[\begin{array}{c} e_{1}\\ e_{2} \end{array}\right]$$

Model 3. Animal model with direct additive genetic effects and common litter effects:$$\left[\begin{array}{c} y_{1}\\ y_{2} \end{array}\right]=\left[\begin{array}{cc} X_{1} & 0\\ 0 & X_{2} \end{array}\right]\left[\begin{array}{c} b_{1}\\ b_{2} \end{array}\right]+\left[\begin{array}{cc} Z_{d1} & 0\\ 0 & Z_{d2} \end{array}\right]\left[\begin{array}{c} a_{d1}\\ a_{d2} \end{array}\right]+\left[\begin{array}{cc} W_{c1} & 0\\ 0 & W_{c2} \end{array}\right]\left[\begin{array}{c} c_{1}\\ c_{2} \end{array}\right]+\left[\begin{array}{c} e_{1}\\ e_{2} \end{array}\right]$$

Model 4. Animal model with direct and maternal genetic effects as well as the common litter effects:$$\left[\begin{array}{c} y_{1}\\ y_{2} \end{array}\right]=\left[\begin{array}{cc} X_{1} & 0\\ 0 & X_{2} \end{array}\right]\left[\begin{array}{c} b_{1}\\ b_{2} \end{array}\right]+\left[\begin{array}{cc} Z_{d1} & 0\\ 0 & Z_{d2} \end{array}\right]\left[\begin{array}{c} a_{d1}\\ a_{d2} \end{array}\right]+\left[\begin{array}{cc} Z_{m1} & 0\\ 0 & Z_{m2} \end{array}\right]\left[\begin{array}{c} a_{m1}\\ a_{m2} \end{array}\right]+\left[\begin{array}{cc} W_{c1} & 0\\ 0 & W_{c2} \end{array}\right]\left[\begin{array}{c} c_{1}\\ c_{2} \end{array}\right]+\left[\begin{array}{c} e_{1}\\ e_{2} \end{array}\right]$$

Model 5. Animal model with direct and maternal genetic effects, and the maternal environmental effects:$$\left[\begin{array}{c} y_{1}\\ y_{2} \end{array}\right]=\left[\begin{array}{cc} X_{1} & 0\\ 0 & X_{2} \end{array}\right]\left[\begin{array}{c} b_{1}\\ b_{2} \end{array}\right]+\left[\begin{array}{cc} Z_{d1} & 0\\ 0 & Z_{d2} \end{array}\right]\left[\begin{array}{c} a_{d1}\\ a_{d2} \end{array}\right]+\left[\begin{array}{cc} Z_{m1} & 0\\ 0 & Z_{m2} \end{array}\right]\left[\begin{array}{c} a_{m1}\\ a_{m2} \end{array}\right]+\left[\begin{array}{cc} W_{me1} & 0\\ 0 & W_{me2} \end{array}\right]\left[\begin{array}{c} {me}_{1}\\ {me}_{2} \end{array}\right]+\left[\begin{array}{c} e_{1}\\ e_{2} \end{array}\right]$$

In all models, data were assumed to be conditionally distributed as a multinormal distribution. For instance, in model 5:$$\left[\begin{array}{c} y_{1}\\ y_{2} \end{array}\right]\left|b,a_{d},a_{m},{me}_{1},R_{0}\right.\sim \;N\left(\left[\begin{array}{cc} X_{1} & 0\\ 0 & X_{2} \end{array}\right]\left[\begin{array}{c} b_{1}\\ b_{2} \end{array}\right]+\left[\begin{array}{cc} Z_{d1} & 0\\ 0 & Z_{d2} \end{array}\right]\left[\begin{array}{c} a_{d1}\\ a_{d2} \end{array}\right]+\left[\begin{array}{cc} Z_{m1} & 0\\ 0 & Z_{m2} \end{array}\right]\left[\begin{array}{c} a_{m1}\\ a_{m2} \end{array}\right]\;+\left[\begin{array}{cc} W_{me1} & 0\\ 0 & W_{me2} \end{array}\right]\left[\begin{array}{c} {me}_{1}\\ {me}_{2} \end{array}\right],\;{I_{n}⊗R_{0}}_{}\right)$$
where$\;y_{1}\;$and $y_{2}\;$are the vectors of the observations for the first (IMF) and second (fatty acid composition) traits sorted by individual and trait within an individual; $b_{1}$ and $b_{2}\;$are the fixed effects vectors; $a_{d1}$ and $a_{d2}$ are the vectors of direct additive genetic effects; $a_{m1}$ and $a_{m2}$ are the maternal genetic effects vectors; ${me}_{1}$ and ${me}_{2}$ are the maternal environmental effects vectors; $R_{0}$ is the residual co(variance) matrix between the two traits; and X*,*
$Z_{d}$, $Z_{m},W_{c}$, $W_{me}$ are the incidence matrices relating the observations to the corresponding effects.

The direct additive genetic effects and the maternal genetic effects had a multinormal distribution:$$\left[\begin{array}{c} a_{d}\\ a_{m} \end{array}\right]\sim N\left(\left[\begin{array}{c} 0\\ 0 \end{array}\right],A⊗G_{0}\right).$$

The maternal environmental effects and residuals ($e_{1}$ and $e_{2}$) were assumed to be distributed as:$$me\sim N\left(0,I_{me}⊗C_{me}\right),$$
$$e\sim N\left(0,I_{n}⊗R_{0}\right),$$
where $G_{0}$ is the 4 × 4 co(variance) matrix of the direct additive and maternal genetic effects, and $C_{me}$ and $R_{0}$ are the 2 × 2 co(variance) matrices of the maternal environmental and residual effects, respectively. ***A*** is the genetic relationship matrix; ***I**_me_*** and $I_{n}\;$are identity matrices of the same order as the number of levels of maternal environmental effects (me) and the number of observations (***n***). Likewise, for model 3 the common litter effects ($c_{1}$ and $c_{2})$ were assumed to be distributed as $c\sim N\left(0,I_{c}⊗C_{c}\right)$, where $C_{c}$ is the the 2 × 2 co(variance) matrix and $I_{c}$ is the identity matrix of the same order as the number of litters.

The univariate models for IMF included the same effects as the bivariate models, but the $G_{0}$ was a 2 × 2 co(variance) matrix of the direct additive genetic and maternal genetic effects, ***C_me,_ C_c_*** and $R_{0}$**_,_** where 1 × 1 co(variance) matrices, and corresponded to the variance of the maternal environmental, litter and residual effects, respectively. Flat priors were assumed for the fixed effects (b) and $R_{0}$, ***C_me_*, *C_c_*** and $G_{0}\;matrices$.

Marginal posterior distributions were estimated using Gibbs sampling with the GIBBS2F90 software [21]. The results were derived from Markov Chain Monte Carlo (MCMC) chains comprising 300,000 iterations. A burn-in period of 20,000 iterations was applied and only 1 in 200 samples was retained for subsequent inference. Subsequently, the POSTGIBBSF90 software [21] was used to estimate the following parameters: posterior means, median, mode, posterior standard deviation, and the highest posterior density region at 95% probability (HPD95%).

### 2.4. Genotype Data

After slaughtering the animals, about 50 g of the obliquus abdominis muscle specimens were used to extract the genomic DNA following a standard protocol [22]. The genotypes used in this study correspond to 76 8th generation dams and 350 of their 9th generation progeny, using the Affymetrix Axiom OrcunSNP Array (Affymetrix Inc., Santa Clara, CA, USA). The SNP array contains 199,692 genetic molecular markers. After quality control by removing SNPs with minor allele frequency (MAF) lower than 0.05, missing genotype rate per marker >5%, missing genotype rate per individual >3% and mapped to sex chromosomes, a total of 349 animals of the 9th generation and 76 dams of the 8th generation with 88,512 SNPs were utilized for GWAS. The missing genotypes were imputed using the software BEAGLE v4.0 [23], saving only the SNPs with imputation quality score $R^{2}>0.75.$ Quality control was carried out using Axiom Analysis Suite v3.0.1.4 and ZANARDI [24].

### 2.5. Maternal Genome-Wide Association and Gene Annotation

The following model named MGWA [20] was fitted to identify and verify the associated maternal genomic regions. MGWA was a Bayesian multiple marker regression model used with a Bayes B model, in which the genotype of the dam is assigned to the phenotypes of her offspring:$$y=1\mu +Xb+\sum_{j=1}^{k}z_{j}\alpha_{j}\delta_{j}+e$$
where y is the vector of offspring’s phenotypes, 1 is a vector of ones, μ is the trait mean, X is a design matrix for the systematic effects, b is a vector of fixed effects (Sex; 2 levels, Month of slaughtering; 5 levels, Parity order; 3 levels), $z_{j}$ is a vector of genotypic covariates for each SNP or locus *j* (0 = heterozygote Aa, 1 = homozygote AA or 2 = homozygote aa reference alleles), k represents the total number of SNPs remaining post-quality control (88,512), $\alpha_{j}$ denotes the random allele substitution effect for SNP j, and its distribution follows a $t_{\nu }(0,\;{\sigma^{2}}_{\alpha })$, with ${\sigma^{2}}_{\alpha }$ obtained as described in [20], using the genetic variance of each trait estimated across all generations of the divergent selection experiment for IMF. $\delta_{j}$ is a random binary variable (0 or 1) denoting the presence ($\delta_{j}$ = 1 with probability 1-π) or the absence ($\delta_{j}$ = 0 with probability π) of SNPs in the model for a given iteration, e is a vector of the residual values distributed as $N(0,I{\sigma^{2}}_{e})$. Therefore, π probability is the proportion of SNPs that have no effect or variance in the model and is a constant calculated as a function of the number of records and the total number of SNPs as 0.9987 [11]. The MGWA Bayes B model was implemented with the GenSel software v4.9 [25]. The posterior distributions of the unknown effects comprised 500,000 iterations, discarding the first 100,000. Afterward, one in 10 samples was saved to avoid the high correlation between consecutive samples and model parameters of the marginal posterior distribution. Inference of the associations between SNPs and phenotypes was based on 1-Mb genomic windows. A total of 1973 genomic windows were analyzed across the 21 autosomes. The relevance of the association was assessed by calculating the posterior mean of the percentage of the genomic variance explained by each window and by calculating the Bayes Factor (BF) for each SNP. A genomic region was considered as associated when a window explained at least 1% of the total genomic variance and had at least one SNP with BF greater than 10 (relevant SNP). Genomic windows accounting for more than 0.5% of the trait’s genomic variance and having SNPs with a BF greater than 10 were also thought to be associated with the trait. The 1% and 0.5% thresholds, respectively, are 20 and 10 times the expected percentage of the genomic variance explained by each genomic window [26].

Candidate genes in the genomic regions associated with the traits of interest were identified using the Ensembl release 99 databases [27]. The reference genome used was the “*Oryctolagus cuniculus*” genome (OryCun 2.0.97). The functional annotation to uncover the biological functions of the annotated genes was performed using the DAVID v6.8 online annotation database [28,29]. Those genes reported to be involved in the physiological processes related to fat deposition, regulation of fat cell differentiation, and fatty acid and lipid metabolism were considered as potential candidate genes.

## 3. Results and Discussion

### 3.1. Quantifying the Importance of Maternal Genetic Effects

#### 3.1.1. Descriptive Statistics

Table 1 shows the main statistical features of IMF and fatty acids. On average, IMF was 1.08 g/100 g of muscle. The percentage of PUFA had the highest average with a value of 39.29%, closely followed by the SFA with a value of 37.07% and finally the MUFA with a value of 23.77%. The most abundant individual fatty acids were linoleic acid (C18:2n6), palmitic acid (C16:0) and oleic acid (C18:1n9) with 26.95%, 26.70% and 20.45%, respectively, followed by stearic acid (C18:0) and arachidonic acid (C20:4n6) with 9.05 and 6.75%, respectively. Myristic acid (C14:0), palmitoleic acid (C16:1), and linolenic acid (C18:3n3) were found in lower percentages of less than 2% each. These results agree with other studies in rabbits [30]. The present results showed a greater percentage of PUFA, C18:2n6, C18:3n3 and C20:4n6 than in pig, beef and sheep, and lower percentages of C14:0 and C18:0 than in beef and veal [31] which explain the high quality of rabbit meat in terms of human nutrition.

#### 3.1.2. Estimate of Genetic, Maternal, and Common Environment Parameters for IMF

Table 2 shows the posterior mean and the highest posterior density region at 95% of the heritability for direct and maternal effects, and the ratio between the litter and environmental effects and the total phenotypic variance for the IMF using five models with increasing complexity. The direct additive heritability was 0.84 when model 1 was applied (only included direct genetic effect). This heritability decreased to 0.54 when direct additive genetic and litter effects were included in model 3 and reached the lowest values for models 2, 4, and 5, which included direct and maternal genetic effects with heritability estimates of 0.44, 0.45 and 0.46, respectively. Similar results were found in Duroc, where estimates of direct heritability for IMF were reduced by approximately 33% when maternal effects were included in the model [32]. The high heritability value found with model 1 can be explained by the overestimation of the direct effect due to the absence of the maternal and the common environmental effects in the model.

A previous study [33] using the same experimental lines and model 3 reported a lower direct heritability value (0.38) compared to our study. It is worth noting that this previous study [33] only analysed data from rabbits within the first three generations. However, in a subsequent study [15], which included data from seven generations, a heritability estimate (0.53) similar to our results was obtained. To our knowledge, no other estimates of IMF heritability have been published in other lines of rabbits. In pigs, with the same model as model 3 of the present study, similar results were found in Duroc [6,13], whereas lower ${h^{2}}_{d}\;$was reported for Large White, at 0.38, Pietrain, at 0.42 [34] and Iberian, at 0.26 [5], and a higher ${h^{2}}_{d}$ for Landrace, at 0.67 [34]. In cattle, lower heritability (0.29) was found for Red Angus [35] and also for Hereford (0.27) with a model similar to model 5 of the present study [36]. The posterior mean of the heritability of the maternal genetic effects (${h^{2}}_{m}$) was lower for models 4 (0.09) and 5 (0.08) than for model 2 (0.22), and the genetic correlations between direct and maternal genetic effects were nearly null in all models (models 2, 4, and 5). This disparity in heritability estimates could be explained by the absence in model 2 of the common litter effect and the maternal environment effect added in models 4 and 5. In Duroc pigs, [32] using a model similar to model 5 of the present study, slightly higher estimates of maternal heritability were reported for IMF (0.13). In the Rotes Höhenvieh cattle breed, a similar ${h^{2}}_{m}$ (0.25) was found by applying a model equal to model 2 [37]. Nevertheless, in this study a negative direct-maternal genetic correlation ranging from −0.18 to −0.49 was found using different models [37].

Regarding the litter effect (*C*^2^), a posterior mean value of 0.18 was found with model 3, which decreased to 0.14 when model 4 including maternal genetic effects was used. This result suggests that the estimate of the litter effect found with model 3 partly includes the maternal genetic effects. The estimates for the litter effect and maternal environmental effect were similar in our study due to the reduced number of litters per dam. Similar results of *C*^2^ were reported in some breeds of pigs, with estimates of 0.14 for Large White, 0.16 for Pietrain, and 0.18 for Duroc [34,38], whereas an important mean value (0.35) was found for Iberian [5]. In contrast, a weak value (0.03) was reported for Landrace [34].

Ignoring the maternal effects leads to an overestimation of the direct heritability (models 1 and 3). Furthermore, the exclusion of the common environmental effects results in an overestimation of maternal and direct heritability (model 2). As stated by [39], in genetic evaluation models, maternal genetic effects should be taken into account; otherwise, direct heritability may be overestimated. Therefore, models 4 and 5 seem to be the most appropriate models. In fact, there is no difference between them due to the reduced number of litters per dam. Hence, we considered model 4 for the estimation of the variance components of the fatty acids.

#### 3.1.3. Estimates of Genetic, Maternal, and Common Environment Parameters for Fatty Acid Composition

Table 3 shows the posterior mean and highest posterior density region at 95% heritability for direct and maternal effects, and the ratio of litter effect to total phenotypic variance for fatty acid composition estimated by model 4. In line with IMF, fatty acids have high to moderate direct heritability. PUFA had higher posterior mean heritability (0.43) than SFA and MUFA (0.35). The largest ${h^{2}}_{d}$ for individual fatty acids were found for C16:0, C14:0 and C18:3n3, at 0.54, 0.50, and 0.50, respectively, followed by C18:2n6, C16:1, and C18:0 with direct heritability of 0.46, 0.45 and 0.40, respectively. C20:4n6 and C18:1n9 showed the lowest *h*^2^_*d*_ (0.38 and 0.33, respectively). Another study with data of seven generations of these same experimental lines, using a model similar to model 3 of our study, reported a similar result of ${h^{2}}_{d}$ for PUFA (0.59), lower for SFA (0.12), and higher for MUFA (0.56) [10].

Regarding individual fatty acids, [10] found similar results for most fatty acids, except for C16:0 and C18:3n3, which reported lower values (0.16 and 0.18, respectively), and for C18:1n9, which reported higher values (0.53). The size of the data and the different models used for fatty acids in our study may explain some of these observed differences. In pigs, in line with our study, moderate to high heritability were found for most fatty acid compositions, but using a model with common litter effect and without maternal genetic effects [6,40,41]. The posterior mean of the heritability of maternal genetic effects (${h^{2}}_{m}$) ranged from 0.10 to 0.46. MUFA had the highest posterior mean of ${h^{2}}_{m}$ (0.46), followed by PUFA (0.17) and SFA with the lowest value of 0.14. For the individual fatty acids, C16:1, C18:3n3 and C18:2n6 showed the greatest ${h^{2}}_{m}$ estimates (0.46, 0.33 and 0.31, respectively). Meanwhile, the other individual fatty acids had heritabilities ranging from 0.10 to 0.20. To our knowledge, no estimates of maternal genetic effects have been previously reported for fatty acids. The posterior mean *C*^2^ did not exceed 0.09 for C18:1n9, MUFA and PUFA, and ranged from 0.11 to 0.18 for the other fatty acids, except for SFA, the only one with *C*^2^ (0.21) higher than for the maternal genetic effect. In Iberian pigs, the common environmental litter effect had an important impact on the fatty acid composition, ranging from 0.06 to 0.53 [5]. However, in Duroc pigs, the *C*^2^ values ranged from 8% (for C18:2n6) to 15% (for C20:4n6) [6]. Nevertheless, the model used in these studies did not include the maternal genetic effect.

#### 3.1.4. Estimates of Direct Additive and Maternal Genetic Correlations between IMF and Fatty Acid Composition, and the Direct-Maternal Genetic Correlations of Fatty Acid Composition

The relationships between IMF and the fatty acid composition are of interest because both affect organoleptic and nutritional properties of meat [7]. Table 4 shows the posterior mean and the highest posterior density region at 95% of direct additive (ρ_d_) and maternal genetic (ρ_m_) correlations between IMF and fatty acid composition and the correlation between direct and maternal genetic effects (ρ_dm_) for fatty acid composition. The posterior mean of direct genetic correlations was strong and positive between IMF and C14:0, C16:1, and SFA, ranging from 0.92 to 0.97, whereas it was strong and negative for C18:0 with an estimate of −0.80. For C18:1n9, C18:3n3 and MUFA, the ρ_d_ was moderate and positive ranging between 0.36 and 0.52. The IMF was negatively correlated with C20:4n6 (−0.63) and with PUFA (−0.51). A previous study reported similar results for C14:0, C16:0, C18:0, C16:1, and C18:3n3, but different estimates for other fatty acid compositions, showing a lower genetic correlation for SFA (0.30), and higher correlations for C18:1, C12:2n6, C20:4n6, MUFA and PUFA compared to the present findings [10]. They used a model similar to our model 3, but their study was carried out with data from eight generations instead of ten. This led to different estimates and wider confidence intervals. Overall, our results showed more robust correlations between intramuscular fat (IMF) and meat fatty acid composition than those observed in previous studies in pigs [41] and cattle [42,43]. The posterior mean of the maternal genetic correlation (ρ_m_) between IMF and its fatty acids was of the same sign as ρ_d_ (except for C16:0 and SFA). C14:0, C16:1, C18:1n9 and MUFA had a relevant ρ_m_ with IMF, ranging from 0.52 to 0.74, whereas C20:4n6 was strongly and negatively correlated with IMF (−0.81). For the other fatty acids, ρ_m_ was weak with a wide HPD95%, providing little information on the true value of the parameter. The genetic correlation between the direct and maternal additive genetic effect (ρ_dm_) within the trait was almost negative, except for the C20:4n6, which showed a wide HPD95%, including the zero. The direct-maternal genetic correlations were important for C16:0, C18:1n9, C18:2n6, SFA, MUFA and PUFA, giving estimates between −0.53 and −0.89. A considerable negative correlation suggests that both direct and maternal genetic effects are influenced by common genetic factors but have opposing influences on the expression of offspring traits. These results are in line with the conclusions of other studies that were reviewed by [44], who reported that direct-maternal genetic correlations showed high and negative estimates in cattle. As stated by [45], a definitive biological rationale for the conflicting interactions between maternal and direct genetic effects on meat quality traits remains elusive. Based on the results from models that included maternal genetic effects (models 2, 4 and 5), it can be concluded that maternal genetic effects play a significant role in shaping the genetic basis of the traits studied and should not be ignored in the model analyses. These findings therefore support performing GWAS analyses to identify genomic regions and potential candidate genes associated with maternal genetic effects related to the traits studied in two rabbit lines under divergent selection for IMF.

### 3.2. Maternal GWAS for IMF and Fatty Acid Composition

There are few studies focusing on conducting GWAS of maternal genetic effects and their relationship to animal phenotype in livestock. To our knowledge, two GWAS studies in dairy cattle investigated the maternal genetic effects on stillbirth and dystocia [46] and body weight at different ages [47]. In the present study, we assigned the genotype of the dam to her offspring and thus performed a GWAS with the phenotype of the offspring and the genotype of the dam using the MGWA model.

Table 5 shows the 1-Mb SNP window regions that accounted for more than 1% of the genomic variance, with the requirement of having SNPs with a BF greater than 10. These particular regions were considered for the identification of potential candidate genes (PCGs). The maternal genomic regions associated with the traits studied were distributed over four rabbit chromosomes (OCU1, OCU5, OCU19 and OCU20). On OCU1, an extended genomic region (35.9–37.3 Mb; three consecutive windows) was associated with PUFA, C16:1n-7, C16:0 and PUFA/SFA ratio, accounting for 3.2%, 9.9%, 12.2% and 13.9% of the genomic variance, respectively (Table 5).

This associated region harbors the potential candidate genes *ADAMTSL1*, *CNTLN*, *BNC2* and *SH3GL2* (Table 6). *SH3GL2* and *CNTLN* (centlein, centrosomal protein; a neighbouring upstream gene of *SH3GL2*) genes are located on porcine chromosome 1 and appear to be related to synaptic vesicle endocytosis and are associated with lipid binding [48]. A previous GWAS study was performed to investigate the relationship between these two genes and obesity in pigs [49].

The genomic regions identified on OCU5, represented by a single genomic window, included genes with major biological functions directly relevant to the traits studied. Consistently, a maternal genomic region of 10.1 to 10.6 Mb on OCU5 was found to be associated with certain fatty acid composition traits. This region explained 3.5%, 5.1% and 4.3% of the genomic variance for C16:0, SFA and PUFA/SFA ratio, respectively. This result would be expected, as the main component of SFA is the palmitic fatty acid (C16:0). Two important putative candidate genes (*RPGRIP1L* and *FTO*) were found in this region; these genes have biological functions directly related to fat metabolism and deposition. The *FTO* (Fat mass and obesity-associated protein) gene plays an important role in adipocyte proliferation, regulation adipocyte differentiation, regulation of white and brown adipocyte differentiation, key regulators of feed intake/appetite and energy homeostasis. The association between the *FTO* gene and obesity in humans has been demonstrated in many populations, and this gene has been identified as an important candidate gene for fat mass, fat-deposition-related traits and obesity [50]. In addition, the *FTO* gene plays an essential role in postnatal growth by influencing milk fat composition and regulating lipid storage [51,52,53]. This result could partially explain the findings of [6], who found important MGE effects on the fatty acid composition of the meat of Duroc pigs. This may be due to their effects on the milk composition of the dams [54]. Other studies in lambs and calves have confirmed that meat fat composition, fat deposition and fatty acid composition of suckling animals are strongly related to the fat composition of the milk they consumed during the lactation period [55,56,57]. Based on the present results, further research is needed to test this hypothesis by evaluating the relationship between the fatty acid composition of the maternal milk fat and that of the offspring’s meat, comparing the two rabbit lines with high and low IMF.

Several studies confirmed that the *FTO* gene affected significantly muscle development and obesity in pigs. A polymorphism in intron 4 of the *FTO* gene (AM931150:g.276T>G) was found to be linked to fat-related traits such as marbling, back fat thickness, back fat between 3rd and 4th last ribs, and IMF content [58,59,60]. Our results are in agreement with those of [52] who reported that, using the candidate gene approach, polymorphism of the *FTO* gene located on OCU5 was significantly associated with growth and meat quality traits in rabbits. In cattle, the *FTO* gene has been associated with growth, carcass, marbling score and traits related to meat quality [61,62,63] as well as fat and protein yield in milk [64]. In addition, a number of human GWAS have identified the *FTO* gene as a primary candidate among the genomic regions that are associated with obesity [65].

The neighboring gene within this genomic region is *RPGRIP1L* (retinitis pigmentosa GTPase regulator-interacting protein-1-like), which is located upstream of the *FTO* gene. *RPGRIP1L* is involved in adipogenesis and insulin-regulated adipocyte metabolism. It encodes a protein with a conserved C2 domain commonly found in calcium-dependent membrane proteins. This protein binds to phospholipids, inositol polyphosphates, and intracellular proteins. Studies have shown that *RPGRIP1L* is associated with hyperphagic obesity, reduced leptin sensitivity and is expressed in adipose tissue. It is also associated with anthropometric and metabolic parameters, including measures of adiposity and affecting milk fat composition [64,66]. Several studies in humans and mice have found that *FTO* and *RPGRIP1L* genes have a co-expression pattern and that the effects of obesity associated with the *FTO* region could be modulated by changes in *RPGRIP1L* expression, leading to downstream effects on leptin signalling [67]. On OCU19, a genomic region (three consecutive windows) at 48.6–50.1 Mb was relevantly associated with IMF and C20:4n-6 fatty acid, accounting for about 8.0% and 1.2% of the genomic variance of the two traits, respectively. This result would be consistent with the high genetic correlation found between IMF and C20:4n-6 fatty acid in rabbits (−0.89) [10]. Of the 41 positionally annotated genes, 30 are known genes and 11 are novel genes identified by their Ensembl gene ID. Some of the existing genes in this region showed functional annotations related to lipid metabolism. In a previous GWAS, [68] found a strong association of the *TANC2* gene with fat content in the dorsal muscle of common carp fish (*Cyprinus carpio*). Another GWAS revealed that *TANC2* was significantly associated with C14:0 fatty acid in Japanese Black cattle [69]. Furthermore, this gene was differentially expressed in pigs with different IMF levels [70]. *ACE* (angiotensin-I-converting enzyme) gene plays a direct role in fatty acid transport, lipid localization and binding, the process of cell proliferation, differentiation, apoptosis and angiogenesis [71]. The association between polymorphism in the *ACE* gene with overweight, obesity and fat distribution in humans has been previously reported [72,73,74]. The *MAP3K3* gene is involved in fatty acid metabolism and regulates the expression of leptin, which is secreted by adipocytes to regulate food intake and energy expenditure and plays a crucial role in maintaining body weight homeostasis [75]. Previous studies have demonstrated that elevated levels of leptin in the bloodstream can suppress fatty acid esterification in mice [76] and skeletal muscle in pigs [77]. Higher leptin expression is expected when a greater number and higher fat content of adipocytes are present [78,79]. Two divergent selection experiments in pigs have revealed significant associations between leptin and the high accumulation of IMF in muscle tissue [75,80]. A GWAS showed that the *TEX2* gene was associated with C18:1n-9 fatty acid in Japanese Black cattle [81]. *PITPNC1* (phosphatidylinositol transfer protein, cytoplasmic 1), also known as the M-rdgB beta gene, is incorporated in lipid metabolism through encoding for a protein of the Nir/rdgB family that is implicated in wide-ranging cellular functions such as regulation of lipid transport, metabolism, and signalling, among others [82]. A study in pigs found a relevant association between the *PITPNC1* gene and the C18:3n-3 content of IMF [83]. The *PRKCA* (protein kinase C, alpha) gene was found to be associated with IMF accumulation in bovine, as it regulates the insulin receptor signaling pathway [84] and is involved in the regulation of IMF deposition and adipose tissue development in domestic yak [85]. In rabbits, prior to this study, GWAS analyses were conducted to investigate direct genetic effects related to IMF [11] and its fatty acid composition [19], using the same rabbit High-IMF and Low-IMF lines. These studies revealed significant genomic regions at OCU1, OCU8 and OCU13 in relation to IMF content. In particular, the most prominent region at 24.9–26.95 Mb on OCU8 accounted for 7.34% of the genomic variance [11]. In terms of fatty acid composition, a genomic region on OCU1 was identified, showing associations with C14:0, C16:0, SFA, and C18:2n6, explaining 3.5%, 11.2%, 11.3%, and 3.2% of the genomic variance, respectively. Furthermore, an additional region on OCU18 was identified, responsible for up to 8% of the genomic variance in the MUFA/SFA ratio. These studies revealed that the identified regions harbored numerous genes associated with IMF and its fatty acid composition (*SCD*, *PLIN2*, *ERLIN1*, and *MTMR2*). Manhattan plots of the IMF, C16:0, SFA, C16:1n-7, PUFA, and PUFA/SFA are shown in Figure 1, Figure 2, Figure 3, Figure 4, Figure 5 and Figure 6, respectively.

## 4. Conclusions

To our knowledge, this is the first study to assess the importance of maternal genetic effects and identify maternal genomic regions for intramuscular fat and intramuscular fatty acid composition traits in rabbits. Our results revealed that maternal genetic effects are not negligible and should be included in the genetic evaluation models for these traits. Ignoring maternal genetic effects led to an overestimation of direct heritability. Furthermore, excluding common environmental effects led to an overestimation of maternal and direct heritability. Therefore, models 4 and 5 seem to be the most appropriate models to estimate the genetic parameters for IMF and its fatty acid composition. The proportion of phenotypic variance explained by maternal genetic effects for IMF ranged from 8 to 22%, depending on the model. Using model 4, important maternal genetic effects for fatty acids ranged from 10% to 46%. The genetic correlation between the direct and maternal genetic effects within the trait was almost negative. It was important for C16:0, C18:1n9, C18:2n6, SFA, MUFA and PUFA, with estimates between −0.53 and −0.89. This suggests that direct and maternal genetic effects have opposing influences on the expression of offspring traits, although they share a common genetic basis. Maternal genetic effects had important influences on IMF and fatty acid composition and are recommended to be included in genetic evaluation models for these traits. A maternal genomic region associated with IMF was detected on OCU19, which contains some relevant candidate genes (*TANC2*, *ACE*, *MAP3K3*, *TEX2* and *PRKCA*). Two other annotated genes (*SH3GL2* and *CNTLN*) were detected in a genomic region on OCU1 associated with C16:0, SFA, C16:1n-7, PUFA, and PUFA/SFA fatty acids. A genomic region on OCU5 was associated with C16:0, SFA, PUFA, and PUFA/SFA fatty acids and included two important candidate genes directly related to lipid metabolism, binding, and obesity (*FTO* and *RPGRIP1L*). Our results confirm the polygenic nature of IMF and fatty acid composition traits, which are controlled by many genes distributed on many chromosomes. Further studies and fine mapping analyses are needed to validate the associations detected by GWAS. In addition, it is important to determine the relationship between the milk fatty acid composition of the dams and the fatty acid composition of the meat of their offspring, comparing the two High-IMF and Low-IMF rabbit lines.

## Figures and Tables

**Figure 1 animals-13-03071-f001:**
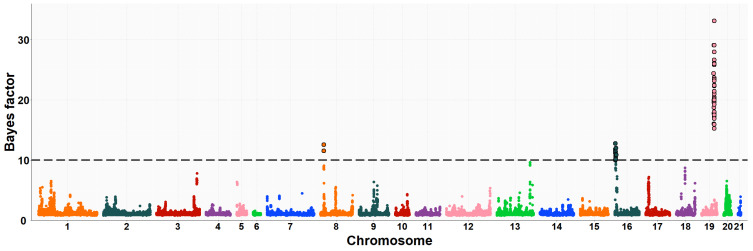
Manhattan plot of the genome-wide association study for intramuscular fat of Longissimus thoracis et lumborum muscle in rabbits using the Bayes factors by each SNP along the rabbit chromosomes under the MGWA model. The Bayes factor threshold of 10 is indicated by the black dashed line.

**Figure 2 animals-13-03071-f002:**
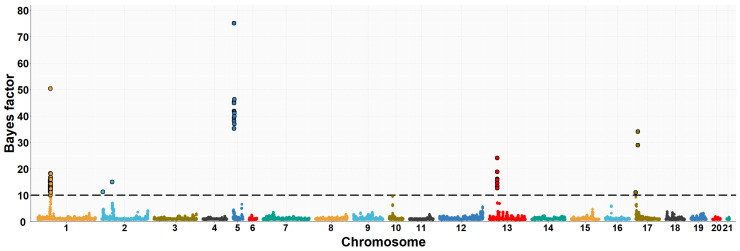
Manhattan plot of the genome-wide association study for palmitic fatty acid (C16:0) of Longissimus thoracis et lumborum muscle in rabbits using the Bayes factors by each SNP along the rabbit chromosomes under the MGWA model. The Bayes factor threshold of 10 is indicated by the black dashed line.

**Figure 3 animals-13-03071-f003:**
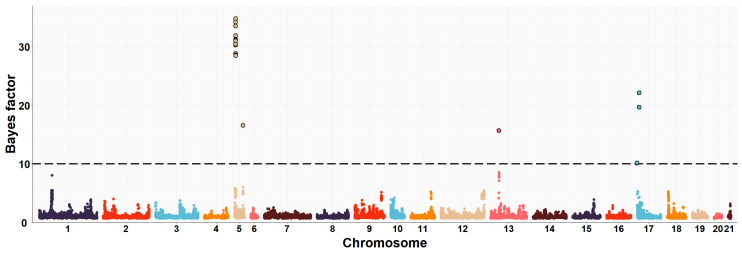
Manhattan plot of the genome-wide association study for saturated fatty acids (SFA) of Longissimus thoracis et lumborum muscle in rabbits using the Bayes factors by each SNP along the rabbit chromosomes under the MGWA model. The Bayes factor threshold of 10 is indicated by the black dashed line.

**Figure 4 animals-13-03071-f004:**
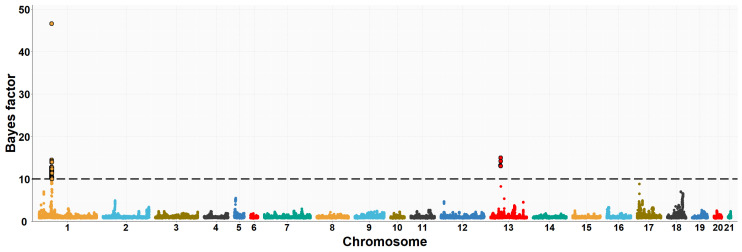
Manhattan plot of the genome-wide association study for palmitoleic fatty acid (C16:1n7) of Longissimus thoracis et lumborum muscle in rabbits using the Bayes factors by each SNP along the rabbit chromosomes under the MGWA model. The Bayes factor threshold of 10 is indicated by the black dashed line.

**Figure 5 animals-13-03071-f005:**
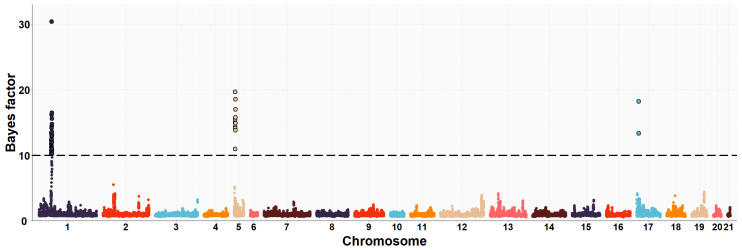
Manhattan plot of the genome-wide association study for polyunsaturated fatty acids (PUFA) of Longissimus thoracis et lumborum muscle in rabbits using the Bayes factors by each SNP along the rabbit chromosomes under the MGWA model. The Bayes factor threshold of 10 is indicated by the black dashed line.

**Figure 6 animals-13-03071-f006:**
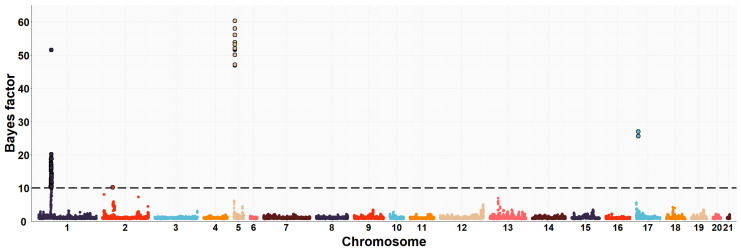
Manhattan plot of the genome-wide association study for the ratio between polyunsaturated and saturated fatty acids (PUFA/SFA) of Longissimus thoracis et lumborum muscle in rabbits using the Bayes factors by each SNP along the rabbit chromosomes under the MGWA model. The Bayes factor threshold of 10 is indicated by the black dashed line.

**Table 1 animals-13-03071-t001:** Mean, median, standard deviation (SD), and coefficient of variation (CV %) for intramuscular fat (g/100 g of muscle) and fatty acid composition of Longissimus thoracis et lumborum muscle (% of total fatty acids).

Trait	Mean	Median	SD	CV (%)
IMF	1.08	1.06	0.23	21.30
C14:0	1.25	1.31	0.52	41.60
C16:0	26.70	26.98	1.57	5.88
C18:0	9.05	9.02	1.26	13.92
SFA	37.07	37.01	1.67	4.50
C16:1n7	1.40	1.35	0.78	55.71
C18:1n9	20.45	20.92	3.26	15.94
MUFA	23.77	24.52	4.55	19.14
C18:2n6	26.95	27.06	2.97	11.02
C18:3n3	1.64	1.69	0.43	26.22
C20:4n6	6.57	6.12	1.99	30.29
PUFA	39.29	38.65	4.86	12.37
MUFA/SFA	0.64	0.66	0.03	4.69
PUFA/SFA	1.06	1.04	0.06	5.66

IMF: intramuscular fat; SFA: saturated fatty acids = C14:0 + C15:0 + C16:0 + C17:0 + C18:0; MUFA: monounsaturated fatty acids = C16:1n7 + C18:1n7 + C18:1n9; PUFA: polyunsaturated fatty acids = C18:2n6 + C18:3n3 + C20:2n6 + C20:3n6 + C20:4n6 + C20:5n3 + C22:4n6 + C22:5n3 + C22:6n3; C14:0: myristic acid; C16:0: palmitic acid; C18:0: stearic acid; C16:1n7: palmitoleic acid; C18:1n9: oleic acid; C18:2n-6: linoleic acid; C18:3n3: α-linolenic acid; C20:4n-6: arachidonic acid. The total number of slaughtered rabbits is 1982.

**Table 2 animals-13-03071-t002:** Marginal posterior mean and highest posterior density region at 95% (between brackets) of the heritability for direct (${h^{2}}_{d}$) and maternal (${h^{2}}_{m}$) genetic effects, the correlation between direct and maternal genetic effects ($\rho_{\mathrm{dm}}$) and the ratio between the litter (***C*^2^**) and maternal environmental (***Me*^2^**) effects and the total phenotypic variance for the Intramuscular Fat of Longissimus thoracis et lumborum muscle using five models.

Trait	${h^{2}}_{d}$	${h^{2}}_{m}$	ρ _dm_	*C* ^2^	*Me* ^2^
Model 1	0.84 [0.69, 0.99]				
Model 2	0.44 [0.24, 0.64]	0.22 [0.08, 0.36]	−0.02 [−0.54, 0.51]		
Model 3	0.54 [0.39, 0.69]			0.18 [0.12, 0.24]	
Model 4	0.45 [0.28, 0.65]	0.09 [0.01, 0.19]	−0.02 [−0.63, 0.59]	0.14 [0.07, 0.21]	
Model 5	0.46 [0.28, 0.63]	0.08 [0.01, 0.19]	0 [−0.64, 0.64]		0.13 [0.06, 0.21]

Model 1: a model fitting the direct additive genetic effects; Model 2 = Model 1 + maternal genetic effects; Model 3 = Model 1 + common litter effects; Model 4 = Model 1 + maternal genetic effects + common litter effects; Model 5 = Model 1 + maternal genetic effects + maternal environmental effects.

**Table 3 animals-13-03071-t003:** Marginal posterior mean and highest posterior density region at 95% (between brackets) of the heritability for direct (${h^{2}}_{d}$) and maternal genetic (${h^{2}}_{m}$) effects and the ratio between the litter effects and the total phenotypic variance (***C*^2^**) for the fatty acid composition of *Longissimus thoracis et lumborum* muscle.

Trait	${h^{2}}_{d}$	${h^{2}}_{m}$	*C* ^2^
C14:0	0.50 [0.34, 0.67]	0.16 [0.07, 0.26]	0.16 [0.09, 0.23]
C16:0	0.54 [0.38, 0.71]	0.15 [0.06, 0.24]	0.14 [0.09, 0.20]
C18:0	0.40 [0.24, 0.56]	0.10 [0.01, 0.19]	0.14 [0.07, 0.20]
SFA	0.35 [0.22, 0.47]	0.14 [0.04, 0.24]	0.21 [0.15, 0.27]
C16:1n7	0.45 [0.27, 0.62]	0.46 [0.38, 0.54]	0.15 [0.08, 0.22]
C18:1n9	0.33 [0.23, 0.43]	0.17 [0.07, 0.26]	0.05 [−0.01, 0.10]
MUFA	0.35 [0.24, 0.46]	0.46 [0.33, 0.60]	0.03 [0, 0.07]
C18:2n6	0.46 [0.35, 0.57]	0.31 [0.18, 0.44]	0.12 [0.05, 0.18]
C18:3n3	0.50 [0.32, 0.67]	0.33 [0.19, 0.47]	0.18 [0.10, 0.26]
C20:4n6	0.38 [0.20, 0.56]	0.20 [0.05, 0.36]	0.11 [0.05, 0.18]
PUFA	0.43 [0.30, 0.55]	0.17 [0.08, 0.27]	0.09 [0.02, 0.15]

Model 4 = direct additive genetic effects + maternal genetic effects + common litter effects; SFA: saturated fatty acids = C14:0 + C15:0 + C16:0 + C17:0 + C18:0; MUFA: monounsaturated fatty acids= C16:1n7 + C18:1n7 + C18:1n9; PUFA: polyunsaturated fatty acids = C18:2n6 + C18:3n3 + C20:2n6 + C20:3n6 + C20:4n6 + C20:5n3 + C22:4n6 + C22:5n3 + C22:6n3; C14:0: myristic acid; C16:0: palmitic acid; C18:0: stearic acid; C16:1n7: palmitoleic acid; C18:1n9: oleic acid; C18:2n-6: linoleic acid; C18:3n3: α-linolenic acid; C20:4n-6: arachidonic acid.

**Table 4 animals-13-03071-t004:** Marginal posterior mean and highest posterior density region at 95% (between brackets) of direct and maternal genetic correlations between intramuscular fat and fatty acid composition and the genetic correlation between additive and maternal genetic effects for fatty acid composition of Longissimus thoracis et lumborum muscle.

Trait	ρ_d_ ^1^	ρ_m_ ^2^	ρ_dm_ ^3^
C14:0	0.96 [0.90, 1.00]	0.68 [0.27, 1.00]	−0.21 [−0.60, 0.18]
C16:0	0.69 [0.46, 0.91]	−0.02 [−0.56, 0.52]	−0.66 [−0.94, −0.39]
C18:0	−0.81 [−0.98, −0.64]	−0.33 [−0.81, 0.16]	−0.15 [−0.61, 0.32]
SFA	0.97 [0.88, 1.00]	−0.03 [−0.55, 0.49]	−0.89 [−1.00, −0.74]
C16:1n7	0.92 [0.81, 1.00]	0.74 [0.50, 0.97]	−0.39 [−0.73, −0.05]
C18:1n9	0.36 [0.15, 0.57]	0.68 [0.44, 0.91]	−0.66 [−0.83, −0.49]
MUFA	0.40 [0.19, 0.61]	0.52 [0.17, 0.88]	−0.53 [−0.69, −0.37]
C18:2n6	−0.27 [−0.51, −0.03]	−0.13 [−0.57, 0.31]	−0.54 [−0.77, −0.31]
C18:3n3	0.52 [0.20, 0.84]	0.33 [−0.39, 1.00]	−0.38 [−0.82, 0.06]
C20:4n6	−0.63 [−0.84, −0.42]	−0.81 [−1.00, −0.61]	0.15 [−0.30, 0.60]
PUFA	−0.51 [−0.69, −0.32]	−0.19 [−0.58, 0.20]	−0.56 [−0.72, −0.40]

^1^ρ**_d_**: genetic correlation of direct additive genetic effects between IMF and fatty acid composition; ^2^ ρ**_m_**: genetic correlation of maternal genetic effects between IMF and fatty acid composition; ^3^ ρ**_dm_**: genetic correlation between direct and maternal additive genetic effects for fatty acid composition; IMF: intramuscular fat; SFA: saturated fatty acids = C14:0 + C15:0 + C16:0 + C17:0 + C18:0; MUFA: monounsaturated fatty acids = C16:1n7 + C18:1n7 + C18:1n9; PUFA: polyunsaturated fatty acids = C18:2n6 + C18:3n3 + C20:2n6 + C20:3n6 + C20:4n6 + C20:5n3 + C22:4n6 + C22:5n3 + C22:6n3; C14:0: myristic acid; C16:0: palmitic acid; C18:0: stearic acid; C16:1n7: palmitoleic acid; C18:1n9: oleic acid; C18:2n-6: linoleic acid; C18:3n3: α-linolenic acid; C20:4n-6: arachidonic acid.

**Table 5 animals-13-03071-t005:** The percentage of genomic variance explained by each relevant genomic region ^1^ associated with intramuscular fat and fatty acid composition of Longissimus thoracis et lumborum muscle in rabbits.

OCU	Map Position (bp)	Trait (% of Explained Variance ^2^)	No. of SNPs ^3^
1	35,902,181–37,321,725	PUFA/SFA (13.9%)	50
1	35,902,181–37,337,488	C16:0 (12.2%)	48
1	35,902,181–37,337,488	C16:1n7 (9.9%)	43
1	35,902,181–37,337,488	PUFA (3.2%)	44
5	10,141,256–10,568,012	C16:0 (3.5%)	12
5	10,141,256–10,568,012	SFA (5.1%)	12
5	10,141,256–10,568,012	PUFA/SFA (4.3%)	12
5	10,161,032–10,568,012	C18:2n6 (3.8%)	8
19	48,610,554–50,061,214	IMF (8.0%)	36
19	49,573,782–49,993,977	C20:4n6 (1.2%)	21
20	16,022,958–16,482,079	C20:4n6 (1.6%)	19

^1^: Genomic windows explaining ≥ 1% of the genetic variance and containing SNPs with a Bayes Factor ≥ 10; OCU: rabbit chromosome; ^2^: percentage of genomic variance explained by the window; IMF: intramuscular fat; C16:0: palmitic acid; SFA: saturated fatty acids; C16:1n7: palmitoleic acid; C18:2n-6: linoleic acid; C20:4n-6: arachidonic acid; PUFA: polyunsaturated fatty acids; ^3^: number of SNPs with a Bayes Factor ≥ 10.

**Table 6 animals-13-03071-t006:** Genomic regions associated with intramuscular fat and fatty acid composition of Longissimus thoracis et lumborum muscle in rabbits.

Trait	OCU	Map Position (bp)		No. of Genes	PCG ^1^
		Start	End		
IMF	19	48,000,588	50,994,791	41	*TANC2, CYB561, ACE, ENSOCUG00000034369, KCNH6, DCAF7, MAP3K3, LIMD2, STRADA, CCDC47, DDX42, FTSJ3, ENSOCUG00000004329, SMARCD2, ENSOCUG00000026494, GHB1, CD79B, SCN4A, ENSOCUG00000035912, ICAM2,* *ENSOCUG00000029995, ENSOCUG00000004362, TEX2, ENSOCUG00000034748, PECAM1, ENSOCUG00000007896, POLG2, DDX5, CEP95, SMURF2, ENSOCUG00000023204, ENSOCUG00000015913, BPTF, NOL11, PITPNC1, PSMD12, HELZ, ENSOCUG00000008850, CACNG4, CACNG5, PRKCA*
C16:0	1	35,072,649	36,992,566	4	*ADAMTSL1, SH3GL2, CNTLN, BNC2*
	5	10,022,607	10,999,540	2	*RPGRIP1L, FTO*
SFA	1	36,002,870	36,992,566	2	*SH3GL2, CNTLN*
	5	10,022,607	10,999,540	2	*RPGRIP1L, FTO*
C16:1n7	1	35,072,649	37,977,724	4	*ADAMTSL1, SH3GL2, CNTLN, BNC2*
PUFA	1	35,072,649	37,977,724	4	*ADAMTSL1, SH3GL2, CNTLN, BNC2*
	5	10,022,607	10,999,540	2	*RPGRIP1L, FTO*
PUFA/SFA	1	35,017,581	37,620,996	4	*ADAMTSL1, SH3GL2, CNTLN, BNC2*
	5	9,928,946	10,444,669	2	*RPGRIP1L, FTO*

OCU, rabbit chromosome; bp, base pair; N, number of genes; PCG, positional candidate gene (only protein-coding genes); ^1^ Novel genes are named according to their Ensembl gene ID and retrieved from DAVID database; IMF, intramuscular fat; C16:0, palmitic acid; SFA, saturated fatty acids; C16:1n-7 acid, palmitoleic; PUFA, polyunsaturated fatty acids.

## Data Availability

The data presented in this study are available upon request to the corresponding author.

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
