# Peer review of "Genome-Wide Association Study of Maternal Genetic Effects on Intramuscular Fat and Fatty Acid Composition in Rabbits"

_animals, 2023, doi:10.3390/ani13193071_

Round 1

Reviewer 1 Report

General Comments

The research presented in this manuscript focuses on a genome-wide association study of maternal genetic effects on intramuscular fat and fatty acid composition in rabbits. The analysis is well-executed, utilizing a robust dataset, and the study design is commendable, providing valuable insights. This manuscript holds promise as a valuable contribution to the field of genetic research in rabbits. However, there are certain areas that should be addressed to enhance the manuscript's potential for publication.

Specific comments:

Point 1: On page 1, in lines 25-27, it is recommended to provide the full abbreviations of the studied traits when initially introducing them. Please include the complete abbreviations for all studied traits either at this point or in the simple summary section.

Point 2: In your abstract, following the description of the results, you omitted a conclusion. Please add a brief conclusion of one or two sentences to the abstract that highlights the primary conclusions or interpretations.

Point 3: Please revise the following instances for consistency with the citation format defined on page 1, line 41: "Maternal effects are defined by Wolf et al. [2] as." This format should be maintained consistently throughout the entire manuscript. Please make the necessary adjustments on page 2, lines 90 and 95; page 3, line 97; page 6, line 250; page 7, line 286; page 9, lines 367 and 369.

Point 4: On page 2, lines 55-57, kindly ensure that you format the references correctly using referencing tools. Please write as follows: “…..nutritional value and health issues [7-9]. The direct genetic control of IMF and its fatty acid composition has been determined in rabbits [10, 11]) and pigs [12-14] by both classical quantitative genetics and genomic wide association studies”.

Point 5: Did the authors conduct population stratification analysis using PCA or MDS? This step is important in GWAS analysis and should be addressed.

Point 6: Include the year of the experiment or the duration of data collection, along with the slaughter year, in the Animals and phenotypes section (2.2).

Point 7: What were the feeding conditions for the experimental animals?

Point 8: On page 2, lines 82-83, could you please provide the rationale for citing references [15] and [16] in this sentence? Kindly consider removing these citations.

Point 9: On page 2, at line 87, please use "24 hours" instead of "24 h."

Point 10: On page 3, at line 104, kindly provide (remove) a rationale for the inclusion of reference [20]. Additionally, at line 105, please remove the extra space before "to estimate."

Point 11: On page 3, line 109, the authors mention the fixed effects of "month (51 levels)”. Could you please provide an explanation of what "month" signifies in the context of the study within the manuscript?

Point 12: Equations should not be in bold throughout the entire manuscript. Please review and make the necessary corrections.

Point 13: On page 4, at line 158, there should be a space after [21].

Point 14: What is the rationale behind employing a SNP call rate threshold of >5% and selecting animals with a call rate >3% as part of the quality control criteria? Could you please provide an explanation for the choice of these specific QC criteria, considering that a substantial number of SNPs and individuals were excluded? It would be helpful if you could also specify the number of removed SNPs and animals under each criterion in the 2.4 Genotype data section.

Point 15: On page 4, line 175, please include the abbreviation "MGWA" along with its respective reference, and format it without bold font.

Point 16: The author referenced fixed effects on page 3, line 109, specifying "month (51 levels), sex (2 levels), parity order (2 levels: 1st parity, ≥ 2nd parity)." However, on page 4, line 181-182, the author mentioned "(Sex; 2 levels, Month; 5 levels, Parity order; 3 levels)." Please clarify these discrepancies.

Point 17: On page 5, at line 186, is it necessary to include references [20] and [25]? Additionally, on the same line, please remove the extra space before "using."

Point 18: On page 5, line 206, the scientific name of the rabbit should be written in italic form as "Oryctolagus cuniculus."

Point 19: On page 5, lines 208, 222, and 224; the citation should be formatted as "… database [27, 28].", “…in rabbits [29].” and “…beef and veal [30],”.

Point 20: Could you please verify that the title of Table 1 aligns correctly with the number of columns in the table? Additionally, consider using "CV (%)" instead of "CV x 100" in the table.

Point 21: Please remove superscript for identifying the words from the entire manuscript Tables like as “IMF” not to use “IMF1

Point 22: On page 6, please move the references [32] and [31] to the end of the sentences in lines 248 and 262, instead of placing them in the middle. Additionally, in lines 253 and 273, rewrite as follows: "found in Duroc [13, 6]" and "for Duroc [33, 37]." On the same page, in line 265, remove the extra space before "Nevertheless."

Point 23: On page 7 and 9, lines 285, 286 and 372-373, ensure that you write the capital and lowercase letters inside the first braces in the same format as the word "Model."

Point 24: Page 8, lines 316, 321, 330, and 332; please rectify the referencing to adhere to the specified format, as previously instructed.

Point 25: On page 9, at line 346, could you please clarify the reasoning behind the inclusion of the sentence and reference "...respectively. [10] using a model"? Kindly reformat this sentence appropriately. Additionally, ensure that the references in lines 351 and 356 are adjusted in accordance with the previous instructions.

Point 26: On page 10, lines 396-397, please eliminate the phrase "(more details in Materials and Methods)." Lines 397-399 have already been addressed in the Materials and Methods section, so kindly remove the statement "Data of ... quality control."

Point 27: On page 10, line 400, the authors have selected more than 1% of the genetic variance. Please include the necessary references to justify this in the Materials and Methods section.

Point 28: Please refrain from using bold formatting for the values in the last column of Table 5.

Point 29: On pages 11, 12, 13, and 15, please format all gene names in italics throughout the entire manuscript.

Point 30: In Table 6 and on page 12, line 475, it is preferable to use the official gene ID names rather than their Ensembl gene IDs for all genes.

Point 31: Page 12, lines 453, 454, 466, 473 - Please revise the references in the appropriate format, as previously instructed. Additionally, on Line 469, please eliminate the space before "...On."

Point 32: On page 13, revise the references in lines 484, 488, 490, 491, 501, 505, and 510 to adhere to the correct format previously specified. Furthermore, in line 487, when discussing "Many investigations...," please include the appropriate references to substantiate this claim, or alternatively, replace it with "Previous investigations..." as suggested.

Point 33: Please include information about the dashed line representing the significant threshold level in the legend for each of Figures 1 through 6.

Point 34: The authors mentioned the Gene Ontology analysis performed using DAVID tools in this research, as stated on line 208 (page 5). However, there are no results and discussions pertaining to this analysis in the manuscript. Please present the Gene Ontology results in a table and elaborate on them in the manuscript, providing explanations for the identified GO terms.

The response concerning the English language is generally acceptable. However, it requires minor corrections for grammatical errors.

Author Response

Response to comments by Reviewers

Review of the Manuscript ID animals-2614705

Title: "Genome-wide association study of maternal genetic effects on intramuscular fat and fatty acid composition in rabbits"

Response to the comments of the Referee

Referee: 1

General Comments

The research presented in this manuscript focuses on a genome-wide association study of maternal genetic effects on intramuscular fat and fatty acid composition in rabbits. The analysis is well-executed, utilizing a robust dataset, and the study design is commendable, providing valuable insights. This manuscript holds promise as a valuable contribution to the field of genetic research in rabbits. However, there are certain areas that should be addressed to enhance the manuscript's potential for publication. AUTHORS:

We would like to appreciate the acknowledgment of the expert reviewer for his/her fruitful comments and suggestions.

Referee: 1

Specific Comments

Point 1: On page 1, in lines 25-27, it is recommended to provide the full abbreviations of the studied traits when initially introducing them. Please include the complete abbreviations for all studied traits either at this point or in the simple summary section.

AUTHORS:

The studied traits have been defined in the Abstract.

Referee: 1

Point 2: In your abstract, following the description of the results, you omitted a conclusion. Please add a brief conclusion of one or two sentences to the abstract that highlights the primary conclusions or interpretations.

AUTHORS:

A conclusion sentence has been added to the Abstract.

Referee: 1

Point 3: Please revise the following instances for consistency with the citation format defined on page 1, line 41: "Maternal effects are defined by Wolf et al. [2] as." This format should be maintained consistently throughout the entire manuscript. Please make the necessary adjustments on page 2, lines 90 and 95; page 3, line 97; page 6, line 250; page 7, line 286; page 9, lines 367 and 369.

AUTHORS:

The definition of the maternal genetic effects was unified throughout the whole manuscript in accordance with the definition of Wolf and Wade (2009).

Referee: 1

Point 4: On page 2, lines 55-57, kindly ensure that you format the references correctly using referencing tools. Please write as follows: “…..nutritional value and health issues [7-9]. The direct genetic control of IMF and its fatty acid composition has been determined in rabbits [10, 11]) and pigs [12-14] by both classical quantitative genetics and genomic wide association studies”.

AUTHORS:

The references formats have been adjusted.

Referee: 1

Point 5: Did the authors conduct population stratification analysis using PCA or MDS? This step is important in GWAS analysis and should be addressed.

AUTHORS:

Prior to the maternal GWAS, the phenotypes were resampled many times to remove any possible relationship between the phenotypes and genotypes, using the following covariates: sex, month of slaughtering, parity order, and the three first components of the principal component analysis of the genotypes in order to avoid population structures. This permutation testing was performed using Plink software.  

Referee: 1

Point 6: Include the year of the experiment or the duration of data collection, along with the slaughter year, in the Animals and phenotypes section (2.2).

AUTHORS:

The following sentence has been added to the section of Animals and phenotypes:

"This experiment spanned ten generations (during the period from 2011 to 2019) and was carried out in a selection nucleus located at the farm of the Institute of Animal Science and Technology at the Universitat Politècnica de València, Valencia, Spain"

* As the selection is based on the offspring phenotypic value of IMF, so, the slaughtering is being performed each generation throughout the experimental period.

Referee: 1

Point 7: What were the feeding conditions for the experimental animals?

AUTHORS:

The information regarding the management and feeding regime of the rabbit lines have been included in the Animals and phenotypes section.

Referee: 1

Point 8: On page 2, lines 82-83, could you please provide the rationale for citing references [15] and [16] in this sentence? Kindly consider removing these citations.

AUTHORS:

These two references were added in order to explain in details the steps and procedure of the divergent selection experiment. The sentence has been modified.  

Referee: 1

Point 9: On page 2, at line 87, please use "24 hours" instead of "24 h."

AUTHORS:

Adjusted

Referee: 1

Point 10: On page 3, at line 104, kindly provide (remove) a rationale for the inclusion of reference [20]. Additionally, at line 105, please remove the extra space before "to estimate."

AUTHORS:

The included reference is important for the readers to describe and explain with details the construction and assumptions of the multivariate Bayesian animal models used to analyze the phenotypic data, especially with the existence of many random effects despite the random additive genetic effect. Additionally, this reference is already cited in the definition of the model terms in the same section" Maternal Genome-wide association and Gene annotation ". The extra space has been removed.

Referee: 1

Point 11: On page 3, line 109, the authors mention the fixed effects of "month (51 levels)”. Could you please provide an explanation of what "month" signifies in the context of the study within the manuscript?

AUTHORS:

Month has been changed to Month of slaughtering in the models of analyses.

Referee: 1

Point 12: Equations should not be in bold throughout the entire manuscript. Please review and make the necessary corrections.

AUTHORS:

All the equations of the models of analyses have been adjusted.

Referee: 1

Point 13: On page 4, at line 158, there should be a space after [21].

AUTHORS:

Corrected

Referee: 1

Point 14: What is the rationale behind employing a SNP call rate threshold of >5% and selecting animals with a call rate >3% as part of the quality control criteria? Could you please provide an explanation for the choice of these specific QC criteria, considering that a substantial number of SNPs and individuals were excluded? It would be helpful if you could also specify the number of removed SNPs and animals under each criterion in the 2.4 Genotype data section.

AUTHORS:

One of the main issues of the rabbit array to be applied to our populations is that this array was built using mostly pet rabbits and without considering our commercial populations. This is the main reason why a huge number of SNPs were removed.

The QC Criteria used for filtering the SNP data are the result of a compromise between the common criteria used in these kinds of studies and the previous exploratory analysis of these data that were previously used (i.e. Laghouaouta et al., Animals 2020).

A SNP call rate threshold of >5% was applied because we would like to be sure that the SNP was not polymorphic by error. The animals with a call rate >3% were used in previous studies and were kept, but no difference between a call rate >3% or >5% was found. In fact, only one sample was removed because of this criterion.

A better explanation of the genotypes used has been included to understand that the main QC filter was done by the SNP call rate threshold.

Referee: 1

Point 15: On page 4, line 175, please include the abbreviation "MGWA" along with its respective reference, and format it without bold font.

AUTHORS:

Adjusted and the respective reference was added.

Referee: 1

Point 16: The author referenced fixed effects on page 3, line 109, specifying "month (51 levels), sex (2 levels), parity order (2 levels: 1st parity, ≥ 2nd parity)." However, on page 4, line 181-182, the author mentioned "(Sex; 2 levels, Month; 5 levels, Parity order; 3 levels)." Please clarify these discrepancies.

AUTHORS:

This is because for the quantification of the maternal genetic effects to assess their importance (quantitative analysis using different Bayesian animal models) all the data set from all the selection generations was used (10 generation of selection; data of 1982 rabbits). However, for the GWAS, only data of only two generations (dams and their offspring); a total of 349 animals of the 9th generation and 76 dams of the 8th generation were used.  

Referee: 1

Point 17: On page 5, at line 186, is it necessary to include references [20] and [25]? Additionally, on the same line, please remove the extra space before "using."

AUTHORS:

You are right; the two references add the same information, so we deleted the ref. [25].

Referee: 1

Point 18: On page 5, line 206, the scientific name of the rabbit should be written in italic form as "Oryctolagus cuniculus."

AUTHORS:

Adjusted

Referee: 1

Point 19: On page 5, lines 208, 222, and 224; the citation should be formatted as "… database [27, 28].", “…in rabbits [29].” and “…beef and veal [30],”.

AUTHORS:

Corrected

Referee: 1

Point 20: Could you please verify that the title of Table 1 aligns correctly with the number of columns in the table? Additionally, consider using "CV (%)" instead of "CV x 100" in the table.

AUTHORS:

Corrected accordingly

Referee: 1

Point 21: Please remove superscript for identifying the words from the entire manuscript Tables like as “IMF” not to use “IMF1”

AUTHORS:

Adjusted in the whole manuscript

Referee: 1

Point 22: On page 6, please move the references [32] and [31] to the end of the sentences in lines 248 and 262, instead of placing them in the middle. Additionally, in lines 253 and 273, rewrite as follows: "found in Duroc [13, 6]" and "for Duroc [33, 37]." On the same page, in line 265, remove the extra space before "Nevertheless."

AUTHORS:

Done

Referee: 1

Point 23: On page 7 and 9, lines 285, 286 and 372-373, ensure that you write the capital and lowercase letters inside the first braces in the same format as the word "Model."

AUTHORS:

Revised

Referee: 1

Point 24: Page 8, lines 316, 321, 330, and 332; please rectify the referencing to adhere to the specified format, as previously instructed.

AUTHORS:

Adjusted

Referee: 1

Point 25: On page 9, at line 346, could you please clarify the reasoning behind the inclusion of the sentence and reference "...respectively. [10] using a model"? Kindly reformat this sentence appropriately. Additionally, ensure that the references in lines 351 and 356 are adjusted in accordance with the previous instructions.

AUTHORS:

The sentence has been rephrased

Referee: 1

Point 26: On page 10, lines 396-397, please eliminate the phrase "(more details in Materials and Methods)." Lines 397-399 have already been addressed in the Materials and Methods section, so kindly remove the statement "Data of ... quality control."

AUTHORS:

Removed

Referee: 1

Point 27: On page 10, line 400, the authors have selected more than 1% of the genetic variance. Please include the necessary references to justify this in the Materials and Methods section.

AUTHORS:

A genomic window was considered to be associated with a trait if it made up more than 1.0% of the trait's genomic variance. In addition, SNPs with a Bayes Factor (BF) of larger than 10 and genomic windows accounting for more than 0.5% of the trait's genomic variance were also thought to be associated with the trait. The 1% and 0.5% thresholds, respectively, are 20 and 10 times the expected percentage of the genomic variance explained by each genomic window. SNPs with BF values above 10 were thought to be associated with the studied trait (Kass and Raftery, 1995). This information was now clearly explained in the M&M.

Referee: 1

Point 28: Please refrain from using bold formatting for the values in the last column of Table 5.

AUTHORS:

Adjusted

Referee: 1

Point 29: On pages 11, 12, 13, and 15, please format all gene names in italics throughout the entire manuscript.

AUTHORS:

All the genes have been formatted in italic in the whole manuscript.

Referee: 1

Point 30: In Table 6 and on page 12, line 475, it is preferable to use the official gene ID names rather than their Ensembl gene IDs for all genes.

AUTHORS:

Since the rabbit is a new species in genotyping and most of its genes are still unknown, the studies concerning genomic characterization in rabbits are scarce, in these studies the genes are nominated as in Table 6; we only nominate the novel genes according to their Ensembl gene ID.

Referee: 1

Point 31: Page 12, lines 453, 454, 466, 473 - Please revise the references in the appropriate format, as previously instructed. Additionally, on Line 469, please eliminate the space before "...On."

AUTHORS:

Adjusted according to the required format, the extra space was deleted.

Referee: 1

Point 32: On page 13, revise the references in lines 484, 488, 490, 491, 501, 505, and 510 to adhere to the correct format previously specified. Furthermore, in line 487, when discussing "Many investigations...," please include the appropriate references to substantiate this claim, or alternatively, replace it with "Previous investigations..." as suggested.

AUTHORS:

Adjusted according to the required format, the sentence has been modified.

Referee: 1

Point 33: Please include information about the dashed line representing the significant threshold level in the legend for each of Figures 1 through 6.

AUTHORS:

The sentence "The Bayes factor threshold of 10 is indicated by the black dashed line" has been added to the figures caption to clarify the Bayes Factor threshold.

 Referee: 1

Point 34: The authors mentioned the Gene Ontology analysis performed using DAVID tools in this research, as stated on line 208 (page 5). However, there are no results and discussions pertaining to this analysis in the manuscript. Please present the Gene Ontology results in a table and elaborate on them in the manuscript, providing explanations for the identified GO terms.

AUTHORS:

We apologize for the confusion. We did not perform a Gene Ontology analysis. We used DAVID to retrieve the biological functions of the genes within the most relevant genomic regions. We are aware that the genes of these genomic regions are not directly identified as being different between populations (H and L), as is the case with differential expression analysis.

Reviewer 2 Report

Nagar et al. conducted the GWAS regarding the maternal genetic effects on intramuscular fat and fatty acid composition in rabbits. The results are interesting and can help us understanding the genomic architecture of these complex traits, especially for the potential maternal effects. I have one main concern that why the genomic information was not used, together with pedigree information, for implementing the ssGBLUP approach, after which the SNP effects could be back-solved for the maternal genetic effects, if I understand correctly! Other comments are listed below:

The Abstract needs to be largely improved by including the related information of sample size, the number of markers genotyped and significant/relevant SNPs, functional implications of candidate genes found, the main conclusion obtained, etc. 

L27: Do the MUFA/SFA and PUFA/SFA mean their ratio? If yes, their statistical descriptions need to be provided in Table 1. 

L28-30: Which traits are the highest and lowest values, respectively? It will help readers better understand the results if you show them here. Furthermore, it needs to indicate that if it is the best model by including the maternal genetic effect for these traits. 

L76-85: I suggest authors to describe this process more clearly. For example, how to rank dams as two rabbits were phenotyped per dam? 

In 2.3: Which criterion you used to select the best one among the five models you have compared. 

L197: Are these 1-Mb genomic regions overlapped or not? If yes, what is the step size? If not, how to avoid the possible bias that may be resulted from the discontinuous segmenting? 

L208-210: How to determine the genes that are involved in these biological functions you mentioned here? 

In Table 1: The sample size and units need to be provided for every trait. 

L288: It is difficult to select the best model just according to the magnitude of estimates of heritability. This point should be aware! 

L290: I suggest to unify the use of Doe and Dam throughout the manuscript. 

In 3.2, All gene names need to be italic. 

L394: Please clarify more about how to assign dam’s genotype to their offspring, and provide the related reference here. 

In Table 5: Please indicate how many relevant SNPs found within each genomic region.

Author Response

Response to comments by Reviewers

Review of the Manuscript ID animals-2614705

Title: "Genome-wide association study of maternal genetic effects on intramuscular fat and fatty acid composition in rabbits"

Response to the comments of the Referee

AUTHORS:

We would like to appreciate the acknowledgment of the expert reviewer for his/her fruitful comments and suggestions.

Referee: 2

Nagar et al. conducted the GWAS regarding the maternal genetic effects on intramuscular fat and fatty acid composition in rabbits. The results are interesting and can help us understanding the genomic architecture of these complex traits, especially for the potential maternal effects. I have one main concern that why the genomic information was not used, together with pedigree information, for implementing the ssGBLUP approach, after which the SNP effects could be back-solved for the maternal genetic effects, if I understand correctly! Other comments are listed below:

AUTHORS:

Numerous methods are available for conducting a Genome-Wide Association Study (GWAS), and we acknowledge the potential of the ssGBLUP approach. Nevertheless, it's worth noting that this method exhibits relatively low shrinkage compared to Bayes B. In situations with limited data, as in our case, this may hinder the identification of the primary genomic regions.

Referee: 2

The Abstract needs to be largely improved by including the related information of sample size, the number of markers genotyped and significant/relevant SNPs, functional implications of candidate genes found, the main conclusion obtained, etc.

AUTHORS:

However, we are restricted by the limited number of words specified by the journal (200 words); we tried to put the most relevant information. The Abstract has been improved.

Referee: 2

L27: Do the MUFA/SFA and PUFA/SFA mean their ratio? If yes, their statistical descriptions need to be provided in Table 1.

AUTHORS:

The statistics regarding MUFA/SFA and PUFA/SFA ratios have been added to Table 1 as recommended.

Referee: 2

L28-30: Which traits are the highest and lowest values, respectively? It will help readers better understand the results if you show them here. Furthermore, it needs to indicate that if it is the best model by including the maternal genetic effect for these traits.

AUTHORS:

The traits of lower and higher maternal heritabilities have been added.

Referee: 2

L76-85: I suggest authors to describe this process more clearly. For example, how to rank dams as two rabbits were phenotyped per dam?

AUTHORS:

In selection experiments for IMF, the data cannot be normally measured on the same individuals that will be used as parents and they have to be measured on relatives, then selection is based on the values of relatives. Selection can be made on the second parities using the IMF value of 2 full sibs (a male and a female) of the first parities. All dams were then ranked according to the IMF values obtained by their offspring. The 20% best dams provided all females for the next generation. Each sire was mated with 5 dams, and 1 male of the best dam was selected for the next generation. This selection within male family was performed to reduce inbreeding. The heritability of IMF (the selection criterion) is high ≈ 0.54; this means that we do not need many offspring to evaluate the dams; in addition, the genetic gain is expected to be high. The existence of two rabbit lines (High-IMF and Low-IMF) facilitates the estimation of the selection response.  

Referee: 2

In 2.3: Which criterion you used to select the best one among the five models you have compared.

AUTHORS:

To compare the five fitted models, we estimate the Deviance Information Criterion (DIC) for each model used, with models 4 and 5 having the lowest DIC values compared to the other models. Although DIC can be a good criterion for overall model fit, it is unclear how much difference in DIC is considered relevant as the number of records and traits included in the model increases. Therefore, we also considered the estimation parameters as an indicator of model fit.  In our study, models that neglected maternal genetic effects overestimated the magnitude of heritability, accompanied by larger standard deviations of the estimates.

Referee: 2

L197: Are these 1-Mb genomic regions overlapped or not? If yes, what is the step size? If not, how to avoid the possible bias that may be resulted from the discontinuous segmenting?

AUTHORS:

The windows were not overlapped for the lack of simplicity. We performed the exploratory analysis using overlapped or not with 10 SNPs size, and no differences were found.

Referee: 2

L208-210: How to determine the genes that are involved in these biological functions you mentioned here?

AUTHORS:

As was previously mentioned. We performed a LD analysis, refined the relevant genomic regions, and identified the genes harboring these. Then, we used the DAVID database and the Gene cards to look at the biological functions of these genes.

Referee: 2

In Table 1: The sample size and units need to be provided for every trait.

AUTHORS:

The sample size is equaled for all traits, as we already mentioned in the materials and methods section (Animals and phenotypes; The present study was carried out with data of 1982 rabbits; out of them 166 were from the base population, 874 from the High-IMF line (H) and 942 from the Low-IMF line (L).). In the footnote of Table 1, we added the total number (The total number of slaughtered rabbits is 1982.). The units are already defined in the Table Title (For Intramuscular Fat (g/100g of muscle), and for fatty acid composition (% of total fatty acids)).

Referee: 2

L288: It is difficult to select the best model just according to the magnitude of estimates of heritability. This point should be aware!

AUTHORS:

As stated before, to compare between the five fitted models, we estimate the Deviance Information Criterion (DIC) for each model used, the model 4 and 5 had the lowest values of DIC compared to the other models. Models were the maternal genetic effects were neglected; the magnitude of the heritability was overestimated accompanied with larger standard deviations of the estimates.

Referee: 2

L290: I suggest to unify the use of Doe and Dam throughout the manuscript.

AUTHORS:

Unified in the whole manuscript

Referee: 2

In 3.2, All gene names need to be italic.

AUTHORS:

All the genes have been formatted in italic in the whole manuscript.

Referee: 2

L394: Please clarify more about how to assign dam’s genotype to their offspring, and provide the related reference here.

AUTHORS:

Through the complete pedigree file, we already knew the dam of the offspring, we already measured the phenotypic data on that offspring, genotyped the dam, so, it was easy using the function merge of R to assign the dam's genotype beside the phenotype of its progeny and by that way we can perform a maternal GWAS. 

Referee: 2

In Table 5: Please indicate how many relevant SNPs are found within each genomic region.

AUTHORS:

The numbers of the relevant SNPs were already included in the table. Nevertheless, the footnotes of the table were modified to clarify this point.